# Smartphone ownership, digital literacy, and the mediating role of social connectedness and loneliness in improving the wellbeing of community-dwelling older adults of low socio-economic status in Singapore

**Amrish Soundararajan**[1,2ᵒ], **Jie Xin Lim**[1,3ᵒ], **Nerice Heng Wen Ngiam**[1,4,5ᵒ], **Angeline Jie-Yin Tey**[1,5,6], **Aaron Kai Wen Tang**[1,5], **Haikel A. Lim**[1,7,8], **Ka Shing Yow**[1,9], **Ling Jie Cheng**[1,10], **Jess Ho**[11], **Qun Xuan Nigel Teo**[5], **Wan Qi Yee**[5], **Sungwon Yoon**[8,12], **Lian Leng Low**[5,12,13], **Kennedy Yao Yi Ng**[1,5,14]*

1 TriGen, Singapore, Singapore, 2 Department of Family Medicine, National University Health System (NUHS), Singapore, Singapore, 3 Wee Kim Kee School of Communication and Information, Nanyang Technological University, Singapore, Singapore, 4 Department of Internal Medicine, Singapore General Hospital, Singapore, Singapore, 5 Population Health and Integrated Care Office, Singapore General Hospital, Singapore, Singapore, 6 Department of Respiratory Care & Critical Care Medicine, Tan Tock Seng Hospital, Singapore, Singapore, 7 Department of Psychiatry, National Health Group, Singapore, Singapore, 8 Health Services and Systems Research, Duke-NUS Medical School, Singapore, Singapore, 9 Department of Internal Medicine, National University Health System, Singapore, Singapore, 10 Health Systems and Behavioral Sciences Domain, Saw See Hock School of Public Health, National University of Singapore, Singapore, Singapore, 11 NTUC Health Co-operative Ltd, Singapore, Singapore, 12 Centre for Population Health Research and Implementation, Singapore Health Services, Singapore, Singapore, 13 SingHealth Duke-NUS Family Medicine Academic Clinical Program, Duke-NUS Medical School, Singapore, Singapore, 14 Division of Medical Oncology, National Cancer Centre Singapore, Singapore, Singapore

ᵒ These authors contributed equally to this work.

* kennedy.ng.y.y@singhealth.com.sg

## Abstract

### Introduction

During the COVID-19 pandemic, safe-distancing measures resulted in many community-dwelling older adults being socially isolated and lonely, with its attending negative impact on wellbeing and quality of life. While digital technology may have mitigated this, older adults of low socioeconomic status (SES) are more likely to be digitally excluded and hence susceptible to the adverse effects of social isolation and loneliness. This study aims to understand the factors that affect digital literacy, smartphone ownership, and willingness to participate in a digital literacy program (DLP), and to test the hypothesized relations between digital literacy, social connectedness, loneliness, wellbeing, and quality of life amongst community dwelling older adults of low SES.

### Materials and methods

A questionnaire assessing digital literacy, social connectedness, wellbeing and quality of life was administered. Socio-demographic variables, pre-existing internet-enabled, and

**Data Availability Statement:** All relevant data are available at DOI: 10.17605/OSF.IO/YVSHJ.

**Funding:** This research is supported by the Singapore Ministry of Health's National Medical Research Council under the Fellowship Programme by SingHealth Regional Health System, Population-based, Unified, Learning System for Enhanced and Sustainable (PULSES) Health Centre Grant (NMRC/CG/C027/2017_SHS), the Healthy, Empowered and Active Living (HEAL) fund and the Infocomm Media Development Authority Digital for Life Fund. There was no other additional external funding received for this study. The funders had no role in study design, data collection and analysis, decision to publish, or preparation of the manuscript.

**Competing interests:** The authors have declared that no competing interests exist.

willingness to participate in a home-based DLP was also collected. Logistic regression was used to identify demographic factors associated with digital literacy, smartphone ownership, and willingness to enroll in a DLP. Serial mediation analysis was also performed using a structural equation model framework.

## Results

A total of 302 participants were recruited. Female gender, older age, lower education levels were associated with lower digital literacy. Those who owned a smartphone tended to be younger and better educated. Older adults who were better educated, of Chinese descent (the ethnic majority in Singapore), and who had lower digital literacy, were most willing to enroll in the digital literacy education program. Social-use digital literacy had a positive indirect effect on well-being ($\hat{b} = 0.037, 95\% \, CI = [0.007, 0.101]$) and Quality of life ($\hat{b} = 0.004, 95\% \, CI = [0.001, 0.010]$), mediated by social connectedness and loneliness. In contrast, instrumental-use digital literacy had a negative indirect effect on well-being ($\hat{b} = -0.030, 95\% \, CI = [-0.080, -0.008]$) and Quality of life ($\hat{b} = -0.003, 95\% \, CI = [-0.008, -0.001]$), mediated by social connectedness and loneliness.

## Discussion

The results suggest there are demographic barriers to participation in DLPs and highlight the benefit of focusing on enhancing social-use digital literacy. Further study is needed to evaluate how well specific interventions to improve social-use digital literacy help to reduce social isolation and loneliness, and ultimately improve wellbeing and quality of life.

## Introduction

Safe distancing measures featured prominently among the public health policies implemented to reduce viral transmission during the COVID-19 pandemic [1, 2]. However, these measures were also associated with adverse consequences such as reducing social connectedness and loneliness [3]. This resulted in detrimental effects on health as social connectedness (objective presence of social contacts) and loneliness (subjective feeling resulting from social isolation) have long been recognized as key social determinants of health [4] such as depression [5–7], low quality of life, lack of wellbeing [8], morbidity [9], and mortality [10]. Social connectedness and loneliness, while distinct, are interrelated [11]. For example, loneliness has been found to mediate the effect of social connectedness on wellbeing [8].

### The role of information and communications technology in reducing social isolation and loneliness

Digital literacy is defined as the ability to use information and communications technology (ICT) to locate, evaluate, use and create information [12]. To overcome the social distance created by safe-distancing measures and the attendant negative consequences like social isolation and loneliness, ICT has proved useful. A pre-pandemic study suggested that ICT use among older adults is associated with fewer depressive symptoms, higher self-rated health, and subjective wellbeing, with these relationships being mediated by reduced loneliness [13]. Hence, an improved digital literacy would enable older adults to extend their social connections and

receive support from these connections [14]; ICT could therefore mitigate the negative effects of emotional and social loneliness caused by the safe-distancing measures implemented during the COVID-19 pandemic [15].

## Addressing the digital divide

Singapore has sped up the implementation of its Digital Transformation Master Plan during the pandemic [16]. The use of ICT, through devices like smartphones, has potential to help older adults maintain social connectedness and reduce loneliness [17]. However, while there are significant benefits to the use of ICT amongst older adults during a pandemic; there undoubtedly remains a significant digital divide with the older adult less likely to access ICT [18]. This issue is multifactorial, ranging from health-related reason [19] to reasons such as negative beliefs or no perceived benefits of ICT, low perceived ability and low digital literacy [18]. Low-income and lower education levels are also identified as key deterrents of ICT usage. Furthermore, the presence of social support to support older adults in their use of ICT seems to be important [20]. As such, older adults of lower socio-economic status (SES) and those with minimal social support are likely to be digitally illiterate and vulnerable to the adverse outcomes of safe-distancing measures during the COVID-19 pandemic, and this may be mediated by greater loneliness and social isolation [21].

## The present study

In response to this digital divide, the Infocomm Media Development Authority Singapore (IMDA) launched the Seniors Go Digital Program, aiming to increase digital literacy among older adults living in Singapore [22]. The program also addresses access issues by providing subsidized smartphones and data subscriptions to seniors of low SES. TriGen, a charity, and Singapore General Hospital, collaborated with IMDA and senior activity centers to bring a home-based digital literacy program (DLP) to these individuals.

This study primarily aims to understand the factors that affect willingness to participate in a DLP. Secondly, this study aims to test the hypothesized relations between digital literacy, social connectedness, loneliness, wellbeing, and quality of life.

As referenced above, ICT use (i.e., digital literacy) helps older adults maintain social connectedness and reduces loneliness [17]. Furthermore, social connectedness and loneliness have been recognized as social determinants of wellbeing and quality of life [9]. Existing literature has also proposed that the impact of social connectedness on wellbeing to be influenced by the mediating factor of loneliness [9]. In light of these findings, Fig 1 presents a conceptual model illustrating the interconnectedness between digital literacy, social connectedness, loneliness, and resulting impact on wellbeing and quality of life.

Given that loneliness and social connectedness are established social determinants of health and well-being, and that greater social connectedness is expected to lower the sense of loneliness, it was hypothesized that the effect of digital literacy on the wellbeing and quality of life of older adults would be serially mediated by social connectedness and loneliness. On a more granular level, it was hypothesized that the mediation effect may be particularly evident for domains of digital literacy pertaining specifically to social interaction.

## Materials and methods

### Participants and procedure

A cross-sectional study was carried out between August 2020 and June 2021. The inclusion criteria for the study were as follows: residents in the South-West region of Singapore; aged >55

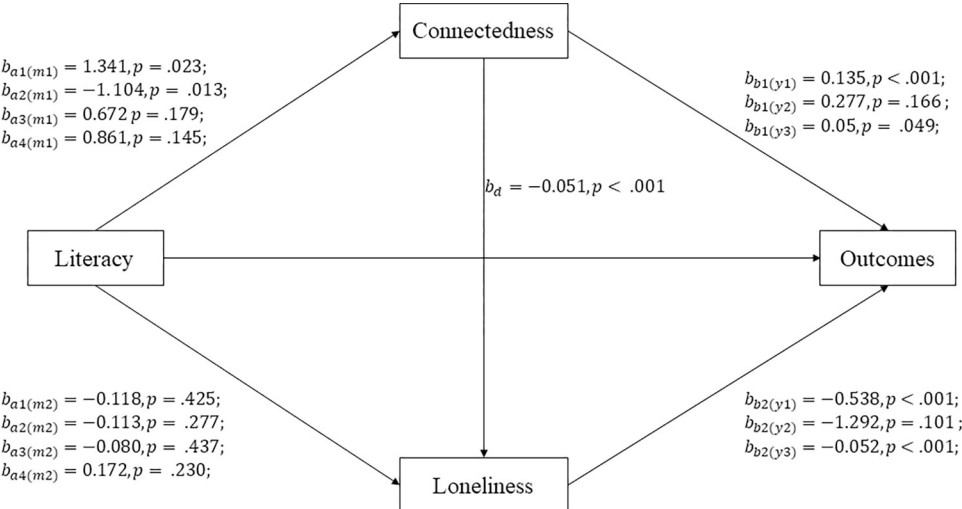

$b_{a1(m1)} = 1.341, p = .023;$
$b_{a2(m1)} = -1.104, p = .013;$
$b_{a3(m1)} = 0.672\ p = .179;$
$b_{a4(m1)} = 0.861, p = .145;$

$b_{b1(y1)} = 0.135, p < .001;$
$b_{b1(y2)} = 0.277, p = .166;$
$b_{b1(y3)} = 0.05, p = .049;$

$b_d = -0.051, p < .001$

$b_{a1(m2)} = -0.118, p = .425;$
$b_{a2(m2)} = -0.113, p = .277;$
$b_{a3(m2)} = -0.080, p = .437;$
$b_{a4(m2)} = 0.172, p = .230;$

$b_{b2(y1)} = -0.538, p < .001;$
$b_{b2(y2)} = -1.292, p = .101;$
$b_{b2(y3)} = -0.052, p < .001;$

**Fig 1. A simplified representation of the serial mediation model and the unstandardized regression weights.**

years; belonging to lower SES (residing in rental public housing, to qualify for which one's gross household income must be less than USD 1100 equivalent [23]) or receiving welfare payments from the Public Assistance Scheme); and who consented to partake in this study (regardless of enrolment in the home-based digital literacy program; DLP). The inclusion criteria of 55 years old and above was determined based on the local context and definition of an older adult [24]. Participants were excluded if they were non-English or Mandarin speaking.

This study utilized convenience sampling: individuals were recruited by working with eldercare and welfare service providers in the community, through phone calls and door-to-door outreach. The team first started off by reaching out to individuals via phone calls. If the individual did not pick up the call despite three attempts made on different days and at different times, at least two physical visits would be conducted (once on a weekday and once on a weekend) to recruit the individual. Failing which, the individual was considered uncontactable.

Written consent was obtained for older adults who were keen to participate in the DLP and verbal consent was obtained for those who declined enrollment in the DLP but still agreed to partake in this research study.

Data was collected from participants either in person or via telephone through standardized questionnaires administered in the participants' preferred language of either English or Mandarin. Standardized training was provided for surveyors prior to household visits for recruitment and questionnaire administration.

Ethical approval was obtained from SingHealth Centralized Institutional Review Board (2020/2722).

## Instruments

Data on socio-demographic variables, pre-existing smartphone ownership and usage was collected through a questionnaire. To measure digital literacy, a questionnaire was constructed to quantify the extent to which the participants use 13 specific functions of an internet-enabled phone (smartphone) based on four aspects of digital literacy: sociability, instrumentality, reassurance, and pastime based on existing literature [25–27]. Refer to S1 Table in S1 Appendix for the questionnaire. Sociability refers to the use of functions or applications for relationship

maintenance; Instrumentality refers to the fulfilment of practical activities of daily life; Reassurance refers to the ability to obtain help in an emergency; and Pastime refers to using the smartphone as an entertainment device [26].

Social connectedness was measured using the Lubben Social Network Scale (LSNS-6), where a total summed score of <12 suggests that participants are at risk of social isolation [28]. Perceived loneliness was assessed using the UCLA 3-item loneliness scale (UCLA-3) [29]; wellbeing was assessed using the Personal Wellbeing Score (PWS) [30] where a higher score indicates better wellbeing. Quality of life was assessed using the EQ-5D-3L and EQ VAS [31]. Value set derived from a local population study was used to calculate a EQ-5D-3L utility index [32]. Details of the study instruments are summarized in S2 Table in S1 Appendix.

## Data analysis

Monte Carlo simulations (Muthén and Muthén's method to determine power [33]) suggested a minimum sample size of 300 was required to achieve a maximum Type I error rate of 0.05 and Type II error rate of 0.20 for most parameters under the assumption of small effect sizes (standardized $\hat{\beta}$ = 0.20) for the serial mediation model.

Principal Component Analysis (PCA) was used to evaluate the composition of the digital literacy index for each of the four aspects of digital literacy. PCA is a common methodology to create composite indices such as the social-economic status [34]. A square matrix of the bivariate Phi coefficient (i.e., a special case of Pearson's correlation coefficient for dichotomous data) of the 13 items (1 = yes, 0 = no) was computed for the analyses using a subset of samples with complete observations (n = 266). PCA was conducted using the pre-determined items for each of the four literacy indices. The four composite indices scores were summed using the dichotomized responses of the respective phone functions. An overall digital literacy index score was computed by summing the four composite indices scores. Scores for social connectedness, loneliness, and wellbeing were computed by summing the item ratings. The EQ-5D-3L utility index was computed using value set derived from the local population [32] as implemented in the eq5d R package [35].

Descriptive statistics and binary logistic regression were conducted using Python 3.7.11 on Google Colaboratory. Univariable analysis was performed to find the associations between socio-demographic variables, digital literacy, smartphone ownership, as well as participants' willingness to enroll in a DLP respectively. Multivariable binary logistic regression models were built on the analyses, to estimate the adjusted effects of selected socio-demographic variables. Per-analysis listwise deletion was utilized for instances of missing data.

The path relationship between digital literacy (sociability, instrumentality, reassurance, and pastime), social connectedness, loneliness, wellbeing, and quality of life was modelled using a serial mediation model in Mplus v8.4 (Fig 1). Full Information Maximum Likelihood was used to accommodate the missing data in the endogenous variables under the missing at random mechanism. A series of simple linear regression models with social connectedness, loneliness, wellbeing, and quality of life as the univariate outcomes regressed on each demographic variable were conducted to identify control variables to be included in the mediation model. Gender was found to be associated with social connectedness and age was found to the associated with quality of life (EQ-5D-3L utility index). These two variables were therefore included in the model to partial out their effect on the respective variable. Age and smartphone ownership were also independently found to be statistically significant predictors of the presence of missing data on the four literacy indices (0 = missing observation, 1 = observation available) using a series of logistic regressions. Hence these two variables were included in the model as auxiliary variables (predictors) of the literacy indices to account for the presence of missing data.

Indirect effects representing the mediation processes were tested using bias-corrected and accelerated (BCa) bootstrap 95% confidence interval with 5000 replications. Standard error of the estimated parameters was also computed using the bootstrapped samples. Model-data fit was evaluated using Hu and Bentler's [36] strategy (i.e., $SRMR \leq 0.08$, $CFI \geq 0.05$, and $RMSEA \leq 0.05$). The nominal $\alpha$ level of 0.05 was used to evaluate all null hypothesis significance testing.

### Reporting

The study was reported following the Strengthening the Reporting of Observational Studies in Epidemiology (STROBE) guidelines for cross-sectional studies.

## Results

### Demographics

A total of 366 older adults were referred. 64 rejected participation in the research or could not be contacted, leaving a total of 302 older adults (82.5%) in the study. The median age is 73, 55% were female, with a representative ethnic mix similar to the general local population [37]. 75.5% had at most 6 years of formal education; 83.4% stayed in public rental housing and 56.6% stayed in one-room apartments (the smallest units in Singapore's public housing developments) suggestive of lower SES. 80.5% of participants already had a mobile phone at the time of the survey (59.9% had smartphones). Almost half (49%) of the participants agreed to enroll in the home-based digital literacy program (DLP; Table 1).

### PCA of digital literacy survey items

PCA was used to evaluate the construction of the literacy indices (sociability, instrumentality, reassurance, and pastime literacy index). Parallel analyses suggested the extraction of one component in each PCA. Subsequent PCA showed that at least 65% of the total variance can be explained by the one-component solution for each of the indices and the component weights were all above 0.60 (see Table 2 for the combined PCA results). The results provided evidence for an equal weighting of the item responses to compute each of the composite index score.

### Digital literacy, loneliness, social connectedness, wellbeing and quality of life

The overall digital literacy amongst the participants was low, with a median score of 3 (observed minimum of 0 and maximum of 13). Participants in general reported low loneliness (median score of 3 on an observed range of 3 to 9) and wellbeing (median score of 8 on an observed range of 0 to 12). 168 (55.6%) of the participants were at risk of social isolation (LSNS6 < 12). The median EQ-5D-3L utility index was 0.854 and the median EQ VAS was 70. Descriptive statistics of the key measurements can be found in S2 Table in S1 Appendix. Participants who owned mobile phones reported better social connectedness (measured using LSNS6) than those without mobile phones (median 11 vs. 8, $p = .002$), as well as better quality of life (median EQ-5D-3L utility index 0.854 vs 0.832, $p = .02$).

### Factors associated with digital literacy, phone ownership and willingness to enroll in a digital literacy program

Comparing participants with an overall digital literacy index $\leq 3$ (i.e., the median) with those with an overall digital literacy index > 3, logistic regression modeling (S3 Table in S1 Appendix)

**Table 1. Demographic characteristics of the participants.**

| Variables | | Values |
|---|---|---|
| **Age (years); median (IQR)** | | 73 (13) |
| **Gender, n (%)** | | |
| | Female | 167 (55.3) |
| | Male | 135 (44.7) |
| **Race, n (%)** | | |
| | Chinese | 234 (77.5) |
| | Malay | 40 (13.2) |
| | Indian | 26 (8.6) |
| | Others | 2 (0.7) |
| **Highest Level of Education[a], n (%)** | | |
| | No formal education | 102 (33.8) |
| | Primary education | 126 (41.7) |
| | Secondary education | 66 (21.9) |
| | Tertiary education | 5 (1.7) |
| **Housing Type[b], n (%)** | | |
| | Rented | 252 (83.4) |
| | Owned (99-year leasehold) | 45 (14.9) |
| **Apartment Size[c], n (%)** | | |
| | One room | 171 (56.6) |
| | Two room | 104 (34.4) |
| | Three room | 21 (7.0) |
| | Four room | 5 (1.7) |
| **Mobile Phone Ownership[d], n (%)** | | |
| | Yes | 243 (80.5) |
| | No | 59 (19.5) |
| **Smartphone Ownership[d], n (%)** | | |
| | Yes | 181 (59.9) |
| | No | 121 (40.1) |
| **Joined home-based digital literacy program, n (%)** | | |
| | Yes | 148 (49.0) |
| | No | 154 (51.0) |

[a] n = 3 (1.0%) were unsure of their education level.

[b] n = 5 (1.7%) were unsure of their housing type.

[c] n = 1 (0.3%) did not indicate the category.

[d] Smartphone is a subset of mobile phone. Mobile phones refer to any types of phones that are not connected to a physical landline, therefore consists of both featurephone and smartphone.

showed that female (compared to male) was independently associated with poorer digital literacy (adjusted odds-ratio [aOR] = 0.49, 95% CI = [0.28. 0.85], $p$ = .011) and was of an older age (aOR = 0.99, 95% CI = [0.98, 1.00], $p$ = .048). Higher education levels (using absence of formal education as the reference group) were independently associated with better digital literacy (Primary education aOR = 2.47, 95% CI = [1.36, 4.49], $p$ = .003; Secondary and Tertiary education aOR = 6.17, 95% CI = [2.98, 12.74], $p < .001$).

Higher education levels also independently increased the odds of smartphone ownership (aOR for Primary education = 2.37, 95% CI = [1.42, 3.98], $p$ = .001; aOR for Secondary and Tertiary education = 5.41, 95% CI = [2.86, 10.62], $p < .001$). Other socio-demographic factors

**Table 2. Results of principal component analyses on the digital literacy survey items[a].**

| Items | Sociability | Instrumentality | Reassurance | Pastime |
|---|---|---|---|---|
| Text Message | 0.85 | | | |
| Video Call | 0.83 | | | |
| Voice Call | 0.75 | | | |
| Government Apps | | 0.90 | | |
| Online Banking | | 0.87 | | |
| Online Purchase | | 0.84 | | |
| Health Apps | | 0.81 | | |
| Read News | | 0.61 | | |
| Call Ambulance | | | 0.97 | |
| Call Police | | | 0.97 | |
| Listen to Music | | | | 0.90 |
| Watch Video | | | | 0.89 |
| Play Games | | | | 0.80 |
| Variance Explained | 66% | 66% | 94% | 74% |

[a] Each column (component) represents results from individual principal component analysis (PCA). Values in the cells represent the PCA component weights.

(gender, race, housing type, apartment size) did not have any statistically significant association, with age being of borderline significance (S4 Table in S1 Appendix).

There was no statistically significant difference in the median ages between those who wished to enroll in the DLP and those who did not. Adjusted using logistic regression, age and gender did not significantly affect the odds of a participant enrolling in a DLP. Instead, higher education levels (compared to no formal education; aOR for Primary education = 1.77, 95% CI = [0.93, 3.36], $p = .081$; aOR for Secondary and Tertiary education = 3.26, 95% CI = [1.44, 7.40], $p = .005$) and prior smartphone ownership (aOR = 4.71, 95% CI = [2.23, 9.94], $p < .001$) independently increased the odds of a participant being willing to enroll in a DLP. Meanwhile a higher digital literacy index (aOR = 0.87, 95% CI = [0.78, 0.96], $p = .002$) and being of the Chinese ethnicity (compared to non-Chinese; aOR = 0.48, 95% CI = [0.24, 0.96], $p = .037$) decreased the odds of being willing to enroll (Table 3).

### Association between digital literacy, social connectedness, loneliness, wellbeing, and quality of life

Serial mediation modeling (SMM) was performed to test the hypothesized association between the digital literary indices, social connectedness ($\alpha = 0.80$), loneliness ($\alpha = 0.87$), wellbeing ($\alpha = 0.86$), and quality of life. Internal consistency (Cronbach's $\alpha$) for the digital literacy scale was not estimated because it was constructed as a formative measure, not a reflective measure. We were unable to estimate the test-retest reliability from this cross-sectional data. For the same reason, estimation of the test-retest reliability of the EQ-5D-3L utility index and EQ VAS (quality of life) was not possible.

SMM decomposes the total effect of a predictor on an outcome into two components: (1) direct effect and (2) indirect effects. For an instance, in this specific model, the total association between social digital literacy on wellbeing can be partitioned into (1) a direct effect of social digital literacy on wellbeing, (2a) a specific indirect effect via social connectedness, (2b) a specific indirect effect via loneliness, and (2c) a specific indirect effect via social connectedness and loneliness. See Fig 1 and Table 4 for a depiction and the results of the serial mediation model.

**Table 3. Factors associated with willingness to join a home-based mobile digital literacy education program.**

| Variables | Willingness to enroll in Digital Literacy Improvement Program | | Univariable Tests | Multivariable Logistic Regression[a] | |
|---|---|---|---|---|---|
| | No (n = 154) | Yes (n = 148) | P-value[b] | Odds Ratio [95% CI] | P-value |
| **Age (year), Median (IQR)** | 73 (13) | 73 (14) | .19** | 0.99 [0.98, 1.00] | .23 |
| **Gender, n (%)** | | | | | |
| Female | 80 (47.9) | 87 (52.1) | .28* | 1.75 [0.96, 3.19] | .067 |
| Male | 74 (54.8) | 61 (45.2) | | -[c] | - |
| **Race, n (%)** | | | | | |
| Chinese | 128 (54.7) | 106 (45.3) | .02* | 0.48 [0.24, 0.96] | .037 |
| Non- Chinese | 26 (38.2) | 42 (61.8) | | -[c] | - |
| **Education Level, n (%)** | | | | | |
| No formal education | 62 (60.8) | 40 (39.2) | .03* | -[c] | - |
| Primary education | 63 (50.0) | 63 (50.0) | | 1.77 [0.93, 3.36] | .081 |
| Secondary and Tertiary education | 29 (40.1) | 42 (59.2) | | 3.26 [1.44, 7.40] | .005 |
| **Housing Type, n (%)** | | | | | |
| Rented | 138 (54.8) | 114 (45.2) | .01* | -[c] | - |
| Owned | 15 (33.3) | 30 (66.7) | | 1.96 [0.90, 4.29] | .09 |
| **Apartment Size, n (%)** | | | | | |
| One room | 92 (53.8) | 79 (46.2) | .31* | - | - |
| Two room | 51 (49.0) | 53 (51.0) | | - | - |
| Larger than Two room | 10 (38.5) | 16 (61.5) | | - | - |
| **Prior smartphone, n (%)** | | | | | |
| No | 78 (65.5) | 41 (34.5) | < .001* | -[c] | - |
| Yes | 75 (41.4) | 106 (58.6) | | 4.71[2.23–9.94] | < .001 |
| **Digital Literacy Index, Median (IQR)** | 3 (7) | 4 (6) | .03** | 0.87 [0.78–0.96] | .005 |

[a] After listwise exclusion of those with missing data, 259 participants were included in the final regression model. The 'apartment size' variable was excluded from the logistic regression model because it is necessarily correlated with 'Housing Type' by definition. In Singapore's public housing scheme, one-room apartments can only be 'rented', whereas two-room and larger apartments can only be 'owned'.

[b] *Chi-squared Test.

**Mann-Whitney U-Test.

[c] Reference group of the variable.

$b_{a1(\cdot)}$ represents the effect of social literacy on the social connectedness (m1) or loneliness (m2), while $b_{a2(\cdot)}$, $b_{a3(\cdot)}$, $b_{a4(\cdot)}$ and represent the effect of instrumental literacy, reassurance literacy, and pastime literacy respectively. Similarly, y1, y2, and y3 in $b_{b1(\cdot)}$ and $b_{b2(\cdot)}$ represents each of the three outcomes: wellbeing, quality of life (EQ-VAS), and quality of life (EQ-5D-3L Utility Index). $\chi^2(22) = 36.333$, $p = .03$, RMSEA = .05, CFI = .99, SRMR = .04.

The serial mediation model results supported the hypothesis that the effect of digital literacy on the wellbeing and quality of life of older adults would be serially mediated by social connectedness and loneliness, with social digital literacy (sociability component) showing the largest positive mediated effect on the outcomes.

The effect of social digital literacy was found to be serially mediated via social connectedness and loneliness ($\hat{\beta} = 0.037$, $95\% \ CI = [0.007, 0.101]$). Referring to Fig 1, this set of results suggested an increase in social digital literacy is associated with an increase wellbeing as a result of the impact of social digital literacy on improving social connectedness ($\hat{\beta}_{a1(m1)} = 1.341$) and the impact of social connectedness on decreasing loneliness ($\hat{\beta}_d = -0.051$), and the negative relationship between loneliness and the reported sense of

**Table 4. Selected results of serial mediation model[a, b].**

| Indirect Effects | $\hat{\beta}$ | 95% CI | $\hat{\beta}_{std}$ |
|---|---|---|---|
| Social -> Social connectedness -> Loneliness -> Wellbeing | <u>0.037</u> | 0.007, 0.101 | 0.013 |
| Social -> Social connectedness -> Loneliness -> Quality of life (VAS) | 0.088 | -0.001, 0.339 | 0.005 |
| Social -> Social connectedness -> Loneliness -> Quality of life (Utility) | <u>0.004</u> | 0.001, 0.010 | 0.014 |
| Instrumentality -> Social connectedness -> Loneliness -> Wellbeing | <u>-0.030</u> | -0.080, -0.008 | -0.013 |
| Instrumentality -> Social connectedness -> Loneliness -> Quality of life (VAS) | -0.073 | -0.278, 0.001 | -0.005 |
| Instrumentality -> Social connectedness -> Loneliness -> Quality of life (Utility) | <u>-0.003</u> | -0.008, -0.001 | -0.013 |
| Reassurance -> Social connectedness ->Loneliness -> Wellbeing | 0.018 | -0.004, 0.062 | 0.006 |
| Reassurance -> Social connectedness ->Loneliness -> Quality of life (VAS) | 0.044 | -0.008, 0.221 | 0.002 |
| Reassurance -> Social connectedness ->Loneliness -> Quality of life (Utility) | 0.002 | 0.000, 0.006 | 0.006 |
| Pastime -> Social connectedness ->Loneliness -> Wellbeing | 0.024 | -0.003, 0.076 | 0.009 |
| Pastime -> Social connectedness ->Loneliness -> Quality of life (VAS) | 0.057 | -0.009, 0.272 | 0.003 |
| Pastime -> Social connectedness ->Loneliness -> Quality of life (Utility) | 0.002 | 0.000, 0.008 | 0.010 |

[a] Wellbeing was measured using the Personal Wellbeing Score (PWS); Quality of life (VAS) was measured using EQ VAS; Quality of life (Utility) was measured using EQ-5D-3L which was converted to the Utility Index; Social connectedness was measured using Lubben Social Network Scale (LSNS6); Loneliness was measured using UCLA3.

[b] Statistically significant (unstandardized) parameter estimate ($\hat{\beta}$) were underlined. Statistical significance of indirect effects was determined using 95% CI. $\hat{\beta}_{std}$ represents the fully standardized parameter estimate.

[c] Refer to S5 Table in S1 Appendix for the full table of the model parameters.

wellbeing ($\hat{\beta}_{b2(y1)} = -0.538$). A positive serial mediation of social connectedness through loneliness on quality of life (EQ-5D-3L utility index) was also found ($\hat{\beta} = 0.004, 95\% \ CI = [0.001, 0.010]$). Fig 1 showed that social digital literacy improved social connectedness which in turn decreased loneliness and a decreased loneliness would subsequently increase the quality of life ($\hat{\beta}_{b2(y3)} = -0.052$).

The effect of instrumental digital literacy (instrumentality component) was also serially mediated via social connectedness through loneliness ($\hat{\beta} = -0.030, 95\% \ CI = [-0.080, -0.008]$). With reference to Fig 1, this set of result suggested that a higher instrumental digital literacy is associated with a lower social connectedness ($\hat{\beta}_{a2(m1)} = -1.104$). Simultaneously, the lowered social connectedness also increased loneliness ($\hat{\beta}_d = -0.051$) which in turn decreased the sense of wellbeing ($\hat{\beta}_{b2(y1)} = -0.538$). A negative serial mediation of social connectedness through loneliness on quality of life (EQ-5D-3L utility index) was also identified ($\hat{\beta} = -0.003, 95\% \ CI = [-0.008, -0.001]$). Referring to Fig 1, this set of results suggested that instrumental digital literacy decreased social connectedness which in turn decreased loneliness and decreasing loneliness would subsequently increase the quality of life ($\hat{\beta}_{b2(y3)} = -0.052$).

In contrast to the impact of social and instrumental digital literacy, reassurance digital literacy and pastime digital literacy did not have any statistically significant association with the wellbeing and quality of life, both directly or indirectly.

# Discussion

The present study sought to underpin factors associated with DLP participation and test the hypothesized relations between digital literacy, social connectedness, loneliness, and wellbeing

and quality of life. The study demonstrated that factors associated with increased willingness to enroll in a DLP included being non-Chinese (ethnic minority), more educated, have a smartphone and poorer self-reported digital literacy. Factors associated with phone ownership included younger age and higher education levels. This corroborates with the existing literature that shows a correlation between sociodemographic factors and digital literacy [18–21].

This study featured a sample of participants coming from a predominantly low SES. More than 50% of the sample was at risk of social isolation, as evidenced through scores on the Lubben Social Network Scale social connectedness measures, significantly lower than a mean LSNS-6 score of 14.16 of a national survey of older Singaporeans conducted in 2016–2017 [19]. Only 59.9% of the samples own an internet-enabled smartphone with a median self-reported digital literacy index of 3. These findings suggest a relationship between social connectedness and social determinants of health, an observation echoed in the existing literature on older adults [8–11].

The results from the serial mediation model demonstrated how better social digital literacy leads to more robust social connectedness, thereby less loneliness, consequently leading to a positive impact on wellbeing and quality of life to a smaller extent. In contrast, there was a negative relationship observed between instrumental digital literacy (the use of applications to perform daily but non-social and non-entertainment related tasks) and wellbeing and quality of life. One plausible explanation for this observation is that social isolation and poorer quality of life necessitates greater independence, fostering greater proficiency in instrumental and reassurance-use functions in an increasingly digital world. Such behaviour can be explained to be driven by an external orientation to the environment motivated by managing external constraints, a contextualism theory advocated by Kraus [38, 39]. Nevertheless, the postulation of a relationship between social isolation and independence warrants further testing via longitudinal research. Furthermore, not all aspects of digital literacy are equally important to wellbeing. Reassurance literacy and pastime literacy were found not to be related to wellbeing and quality of life. The serial mediation model has provided unified evidence of a plausible causal pathway from digital literacy to wellbeing and quality of life through social connectedness and loneliness, which has been evaluated piecewise in the literature.

Findings from this study have implications on the considerations of implementing programs to improve digital literacy in older adults. Lower socioeconomic status, lower education levels and older age are barriers to DLP uptake and participation, so additional efforts will be required to engage these groups to prevent digital exclusion. Whilst designing a policy aimed at increasing digital literacy, one should be cognizant that not all aspects of digital literacy are equally important to wellbeing and quality of life. Our study showed that by and large, social digital literacy is associated with increased social connectedness, reduced loneliness, and consequently, improved wellbeing and quality of life. Policymakers and organizers of future DLPs should thus design programs that are more focused on such specific aspects of digital literacy.

Several methodological limitations exist within the present study. Firstly, given the convenient sampling of the current samples, generalizability of the findings to other samples of the same population of older adults of low SES will have to be tested with independent studies. Secondly, our approach to recruit samples by working with eldercare and welfare service providers may result in clustered samples (i.e., samples nested in service centers) which would result in inaccurate statistical estimates if the presence of clustering was not modelled. However, because the potential clustering units (i.e. the identity of whichever service center referred the respective study subject to us) were not recorded in the de-identified dataset, it was not possible to empirically test for the presence of clustering in the analyses, and thus this limitation must be acknowledged. Thirdly, while it is tempting to interpret the associations found in the mediation model as inherently causal; readers should note that the data in this study was

collected from a cross-sectional study, therefore longitudinal studies need to be conducted to establish the temporal causal relationship.

In conclusion, this study provides an estimate of key factors to partake in DLP and the prevalent patterns of mobile phone ownership, usage and digital literacy of a target demographic that is particularly at risk of getting left behind by digital transformation, and at risk of social isolation. Interventions to increase digital literacy (especially in the social domain) may help to reduce social isolation, loneliness and improve well-being. Such an impact should continue to motivate further efforts to close the digital divide, and particularly to try to reach out to the few who are most difficult to reach but who are most at risk.

## Supporting information

**S1 Appendix. Supplementary tables.**
(DOCX)

## Acknowledgments

The authors wish to thank Ming Hao Wee, Wee Teck Loh from NTUC Health Co-operative Ltd; Montfort Care; Infocomm Media Development Authority; Singapore General Hospital Community Nursing and Hospital-to-Home Program; Elizabeth Puay Ting Pang, Jamica Pei Ying Tan from Singapore General Hospital Population Health and Integrated Care Office, and all of the study's participants. This study would not be possible without the support of these individuals and organizations. We would also like to thank one of the paper's reviewers Dr Senthil Amudhan for pointing the potential methodological limitation of clustering.

## Author Contributions

**Conceptualization:** Nerice Heng Wen Ngiam, Ka Shing Yow.

**Formal analysis:** Amrish Soundararajan, Jie Xin Lim, Qun Xuan Nigel Teo.

**Investigation:** Nerice Heng Wen Ngiam, Haikel A. Lim, Ka Shing Yow, Ling Jie Cheng.

**Methodology:** Nerice Heng Wen Ngiam.

**Project administration:** Nerice Heng Wen Ngiam, Angeline Jie-Yin Tey, Aaron Kai Wen Tang, Ka Shing Yow, Jess Ho, Qun Xuan Nigel Teo, Wan Qi Yee.

**Supervision:** Sungwon Yoon, Lian Leng Low, Kennedy Yao Yi Ng.

**Writing – original draft:** Amrish Soundararajan, Jie Xin Lim, Nerice Heng Wen Ngiam.

**Writing – review & editing:** Amrish Soundararajan, Jie Xin Lim, Nerice Heng Wen Ngiam.

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
