## [Decision Letter · Decision Letter 0]

30 May 2023

PONE-D-23-04295Smartphone Ownership, Digital Literacy, and the Mediating Role of Connectedness and Loneliness in Improving the Wellbeing of Community-Dwelling Older Adults of Low Socio-Economic Status in SingaporePLOS ONE

Dear Dr. Amrish Soundararajan,

Thank you for submitting your manuscript to PLOS ONE. After careful consideration, we feel that it has merit but does not fully meet PLOS ONE’s publication criteria as it currently stands. Therefore, we invite you to submit a revised version of the manuscript that addresses the points raised during the review process.Please address all the suggestions, comments and feedback from the reviewers. Please ensure that your decision is justified on PLOS ONE’s publication criteria and not, for example, on novelty or perceived impact.

We look forward to receiving your revised manuscript.

Kind regards,

Kshitij Karki, MPH, MA

Academic Editor

PLOS ONE

Journal Requirements:

“This research is supported by the Singapore Ministry of Health’s National Medical Research Council under the Fellowship Programme by SingHealth Regional Health System, Population-based, Unified, Learning System for Enhanced and Sustainable (PULSES) Health Centre Grant (NMRC/CG/C027/2017_SHS), the Healthy, Empowered and Active Living (HEAL) fund and the Infocomm Media Development Authority Digital for Life Fund. The funders had no role in study design, data collection and analysis, decision to publish, or preparation of the manuscript.”

3. Please include your full ethics statement in the ‘Methods’ section of your manuscript file. In your statement, please include the full name of the IRB or ethics committee who approved or waived your study, as well as whether or not you obtained informed written or verbal consent. If consent was waived for your study, please include this information in your statement as well

Additional Editor Comments (if provided):

Please address all the comments and feedback from the reviewers.

Reviewers' comments:

Reviewer's Responses to Questions

**Comments to the Author**

1. Is the manuscript technically sound, and do the data support the conclusions?

Reviewer #1: Yes

Reviewer #2: Yes

2. Has the statistical analysis been performed appropriately and rigorously? 

Reviewer #1: Yes

Reviewer #2: Yes

3. Have the authors made all data underlying the findings in their manuscript fully available?

Reviewer #1: Yes

Reviewer #2: Yes

4. Is the manuscript presented in an intelligible fashion and written in standard English?

Reviewer #1: Yes

Reviewer #2: Yes

5. Review Comments to the Author

Reviewer #1: This research is very well done. My comments are seeking for consistency between the results (text) and the values given in Figure 1. Please see comment bubbles in the attachment. Please clarify the values, if needed.

Reviewer #2: The study seeks to explore the connection between digital literacy and various factors such as social connectedness, loneliness, wellbeing, and quality of life (QoL) through mediation analysis. Extensive research has demonstrated the significant influence of social isolation and loneliness on the overall health, mental well-being, and longevity of older individuals, as well as their quality of life. To address these issues, numerous interventions, both face-to-face and digital, have been developed to mitigate social isolation and loneliness among older adults. Given this context, the current manuscript is intriguing and appears to contribute to our existing knowledge by examining the role of digital literacy as a potential mediator in enhancing the well-being of vulnerable elderly individuals.

However, I have the following concerns that needs to be addressed for better clarity and further improvement:

Title: Title needs to be clear, specific and reflect the content and context of the manuscript. It should clearly delineate exposure, outcome, and mediator variables. I presume it is “social connectedness” not “connectedness”

Abstract appears too brief to provide a clear picture for the readers. The abstract can be expanded to better reflect the manuscript's content and context. Information on analytic approach used for mediation analyses and results (including point estimates and uncertainty estimates) should be provided.

Introduction: The description of social connectedness in the opening statement does not seem valid (?? Absence).

For better clarity and flow, the introduction needs to provide adequate information on the vulnerability of elderly for social isolation and loneliness and its consequences before describing Digital Transformation Master Plan.

The introduction can include a brief description of digital literacy in a line or two.

As “propensity” has statistical interpretation; the authors can use appropriate alternate term for “propensity”.

The supporting evidence for why the mediators might affect the outcomes (wellbeing and quality of life of older adults) needs to be strengthened.

Overall, the introduction provides a good description of the study background and theoretical rationale for investigating the mechanisms of interest. The theory that underpins the proposed mechanisms of interest and why the exposure or intervention is expected to affect the proposed mediator (action theory), and why the mediator is expected to affect the outcome (conceptual theory) is also stated explicitly.

Materials & Methods: The statement “The study was reported following the Strengthening the Reporting of Observational Studies in Epidemiology (STROBE) guidelines for cross-sectional studies” appears irrelevant. Methods should focus on how it was conducted not how it was reported.

The authors can mention the rationale for including those aged >55 years as the definition of elderly varies across the countries.

Exclusion criteria needs to be mentioned explicitly. For example, one is not sure about participants who were already enrolled and participated in DLP.

Sampling strategy needs expansion for better clarity. It is possible that different strategies “phone calls and door-to-door outreach” would have introduce selection bias by deviating from randomness.

Reason for differential consent procedure for those who participate and those who do not participate in DLP and its ethical validity.

Possible measurement bias due to different data collection strategies should be mentioned in discussion section.

Languages used for questionnaire administration needs to be mentioned.

It should be “questionnaire administration” not “survey administration”

Though the authors have used the term “digital literacy”, it was smartphone literacy that was specifically addressed. This needs to be reflected in the title and discussion.

For all the study instruments used, scoring pattern, maximum and minimum possible score range and basic psychometric proprieties should be mentioned.

From the description, it was not clear how information on socio-demographic variables, smartphone ownership, as well as participants’ propensity to enroll in a DLP was collected.

Data analysis: The reference for the formula used to derive the minimum sample size estimation based on Monte Carlo simulations should be provided.

The authors should reconfirm whether it was bivariate or Univariable analysis that was performed to find the associations.

Specific assumptions about the causal model (effect modification, positivity, and consistency) , if any can be elaborated.

Also, assumptions on assumptions of multivariate normality and linearity (if any), and post-hoc modifications (if any) can be mentioned. The authors can mention the approach (if any) that was used to explore violation of assumptions with regard to residual confounding, direction of causal relationships and absence of common causes of multiple mediators. This will ensure more robustness and reliability of the analysis.

Authors can mention whether there was clustering due to their sampling strategy; if there was clustering, should describe how clustering was accounted for with regard to within- and between-cluster heterogeneity, and possible effects by participation in DLP programme for the estimation of direct and indirect effects.

Results : The authors should mention the difference between mobile and smartphone and the relevance of describing mobile phone with regard to the stated objectives and included key concepts.

As table 3 reflects the primary objective, the logistic regression model should include all the variables and provide both adjusted and unadjusted odds ratio for all the variables.

The authors should reorganise the mediation results as per the stated hypothesis for better clarity.

Discussion: Overall, the discussion appears weak, terse and needs detailed elaboration by comparing with existing evidence for the stated objectives and hypothesis.

The discussion can highlight how the result findings concord with the stated objectives and hypothesis.

Authors can state any limitations regarding unmeasured confounding, measurement error, model misspecification, selection bias, and missing data.

The authors can bring out the chain mediating effect and its implications in their discussion section.

Authors should add a para on implications of the study findings.

Further, the authors should revisit the results section and shift that information which were not discussed into the supplementary appendix.

6. PLOS authors have the option to publish the peer review history of their article (what does this mean?). If published, this will include your full peer review and any attached files.

Reviewer #1: No

Reviewer #2: **Yes: **Senthil Amudhan

---

## [Author Response · Author response to Decision Letter 0]

18 Jun 2023

Please find attached our responses to the reviewers comments in the attached document, titled "Response to Reviewers". Please also find attached the revised Manuscript, the Revised Manuscript with Tracked Changes, revised Appendix and revised Figure 1.

---

## [Decision Letter · Decision Letter 1]

3 Jul 2023

PONE-D-23-04295R1Smartphone Ownership, Digital Literacy, and the Mediating Role of Social Connectedness and Loneliness in Improving the Wellbeing of Community-Dwelling Older Adults of Low Socio-Economic Status in SingaporePLOS ONE

Dear Dr. Soundararajan,

Thank you for submitting your manuscript to PLOS ONE. After careful consideration, we feel that it has merit but does not fully meet PLOS ONE’s publication criteria as it currently stands. Therefore, we invite you to submit a revised version of the manuscript that addresses the points raised during the review process.

We look forward to receiving your revised manuscript.

Kind regards,

Kshitij Karki, MPH, MA

Academic Editor

PLOS ONE

Journal Requirements:

Additional Editor Comments:

Please address the comment of the reviewer.

Reviewers' comments:

Reviewer's Responses to Questions

**Comments to the Author**

1. If the authors have adequately addressed your comments raised in a previous round of review and you feel that this manuscript is now acceptable for publication, you may indicate that here to bypass the “Comments to the Author” section, enter your conflict of interest statement in the “Confidential to Editor” section, and submit your "Accept" recommendation.

Reviewer #1: All comments have been addressed

Reviewer #2: (No Response)

2. Is the manuscript technically sound, and do the data support the conclusions?

Reviewer #1: Yes

Reviewer #2: Yes

3. Has the statistical analysis been performed appropriately and rigorously? 

Reviewer #1: Yes

Reviewer #2: Yes

4. Have the authors made all data underlying the findings in their manuscript fully available?

Reviewer #1: Yes

Reviewer #2: Yes

5. Is the manuscript presented in an intelligible fashion and written in standard English?

Reviewer #1: Yes

Reviewer #2: Yes

6. Review Comments to the Author

Reviewer #1: (No Response)

Reviewer #2: The authors have addressed most of the concerns diligently.

The authors justification for keeping the term “digital literacy” instead of changing it to “smartphone literacy” is acceptable.

However, the authors response that they did not expect clustering due to our sampling requires more objective evidence. One can find the clustering at the institution level from where the participants were sampled. In this context, the authors need to provide objective evidence that there is no clustering in their data. Consequently, any finding related to clustering should be discussed in the manuscript.

Similarly, the authors aim to create parsimonious logistic regression model need strong justification. With the exclusion of age and gender which are considered to be universal confounders, the current model falls short to provide a robust and valid estimates (The current model wrongly rejects potentially important variables when the relationship between an outcome and a risk factor is confounded and through exclusion when this confounder is not properly controlled). There is possibility that inclusion of age and gender might turn some of the significant variables to non-significant ones.

Rather than parsimonious, the logistic regression model should be robust and more valid. The most unpleasant side effect of variable selection is its impact on inference about true values of regression coefficients by means of tests and confidence intervals. Further, using a threshold P-value <0.05 for variable selection often led to important adjustment variables being dropped from a model due to stochastic variability. Hence, if the authors still want to proceed with variable selection method, they should use the recommended less stringent threshold (P-value <0.25) for variable selection and provide evidence on selection stability and model uncertainty through sensitivity analyses (https://onlinelibrary.wiley.com/doi/epdf/10.1111/tri.12895).

7. PLOS authors have the option to publish the peer review history of their article (what does this mean?). If published, this will include your full peer review and any attached files.

Reviewer #1: No

Reviewer #2: **Yes: **Dr. Senthil Amudhan

---

## [Author Response · Author response to Decision Letter 1]

8 Aug 2023

Please see document attached: Response to Reviewers

---

## [Editor Report · Decision Letter 2]

10 Aug 2023

Smartphone Ownership, Digital Literacy, and the Mediating Role of Social Connectedness and Loneliness in Improving the Wellbeing of Community-Dwelling Older Adults of Low Socio-Economic Status in Singapore

PONE-D-23-04295R2

Dear Dr. Amrish Soundararajan,

We’re pleased to inform you that your manuscript has been judged scientifically suitable for publication and will be formally accepted for publication once it meets all outstanding technical requirements.

Kind regards,

Kshitij Karki, MPH, MA

Academic Editor

PLOS ONE
---

## [Editor Report · Acceptance letter]

21 Aug 2023

PONE-D-23-04295R2 

Smartphone Ownership, Digital Literacy, and the Mediating Role of Social Connectedness and Loneliness in Improving the Wellbeing of Community-Dwelling Older Adults of Low Socio-Economic Status in Singapore 

Dear Dr. Soundararajan:

I'm pleased to inform you that your manuscript has been deemed suitable for publication in PLOS ONE. Congratulations! Your manuscript is now with our production department. 

Kind regards, 

on behalf of

Dr. Kshitij Karki 

Academic Editor

PLOS ONE